# Real-Time Control Technology for a Bio-Liquor Circulation System in a Swine Barn with Slurry Pit: Pilot Scale Study

**DOI:** 10.3390/ani12212941

**Published:** 2022-10-26

**Authors:** Seungsoo Kim, Soomin Shim, Seunggun Won, Junghoon Kwag, Changsix Ra

**Affiliations:** 1Department of Animal Industry Convergence, College of Animal Life Sciences, Kangwon National University, Chuncheon 24341, Korea; 2Department of Animal Resources, College of Life and Environmental Science, Daegu University, Gyeongsan 38453, Korea

**Keywords:** real-time control, bio-liquor, swine wastewater, pilot scale, nitrate knee point, nitrogen break point

## Abstract

**Simple Summary:**

The high intensity of odor emitted from swine production facilities is the biggest complaint from local residents and hinders the development of the swine industry. Furthermore, the odor of swine farms is a core concern of the global swine industry, and various efforts have been made to develop odor reduction technology. Although the bio-liquor circulation system (BCS) as an odor reduction technology has been popular among Korean farmers, the odor reduction effect of BCS has often not been fully obtained because of the lack of an appropriate operating strategy, dependent on each different condition of individual farms. To overcome this limitation, in this study, the bio-reactor was operated in a real-time manner based on a diagnosis of ORP and pH(mV) time-profiles in a pilot scale. From the results of this study, the BCS control strategy using the time-profile of ORP and pH (mV) showed a successful performance for detecting the biochemical changes of nitrogen, achieving self-operation of the bio-reactor and effective odor reduction.

**Abstract:**

The livestock industry, especially swine production, has been pressurized by vicinity complaints about odor in Korea. Therefore, a lot of effort has been undertaken regarding reducing the odor emissions from pigsties, widely carried out and the washing out manure in slurry pit by liquid-phase compost has particularly been spotlighted with outstanding performance of odor reduction. However, such a washing out manure called bio-liquor circulation system (BCS) has been controlled by a timer with designated reaction time, which cannot guarantee the system performance. This research proposes an effective real-time control technology for BCS, which circulates bio-liquor to the slurry pit of swine barns. The real-time control system was operated through accurate detection of the designated control points on the oxidation reduction potential (ORP) and pH time profiles for the nitrate knee point (NKP) and nitrogen break point (NBP) in anoxic and aerobic conditions with 100 and 99.6% performances, respectively. The duration of the anoxic and aerobic phases was also automated and noticeably lowered the concentration of nutrients in the manure in the slurry-pit, which served as a source of malodor. The real-time control strategy may be an innovative way to reduce odor and simultaneously produce liquid fertilizer, and provides a reference for the optimization of the industrial scale.

## 1. Introduction

The increasing global population and changes in dietary habits have led to a rise in livestock farming worldwide. In 2018, 781 million hogs were raised globally and 11.2 million hogs were reared in South Korea [1]. The annual production of swine manure in South Korea exceeds 20.5 million tons, and accounts for 41% of the total domestic annual manure generation [2]. Manure has been used as a fertilizer for centuries and is regarded as a valuable resource for increasing soil productivity [3,4]. Livestock manure contains high nitrogen (N), phosphorus (P), and other nutrient (complex organic compounds, undigested food components, and gut microorganisms) contents. To prevent soil and groundwater contamination, manure needs to be properly treated. Excess N and P cannot be used by plants and is leached into water bodies as a result of surface runoff, which may cause the depletion of dissolved oxygen, eutrophication, and algal blooms [4,5,6,7,8]. Moreover, the landfilling of swine manure and organic waste without proper treatment have been forbidden in Korea since 2005 [9]. Therefore, certain advanced manure management/treatment strategies are required in order to ensure the safety of the manure and that its final disposal is not harmful to ecosystems.

There are four main types of manure management: deep-pit, pull-plug, bedding, and solid−liquid separation [10]. Deep-pit and pull-plug are liquid systems where manure is stored in the pit from two weeks to several months, and then it is decanted before the rearing of a new batch of pigs starts. The bedding and separation systems are solid manure systems and the manure in the bedding system is mixed with bedding materials such as straw, rice husk, and sawdust during the in-house period. The mixture of manure and bedding is subjected to either stockpiling or composting before land application. In the separation system, solids and liquids are separated, and solids are removed from the swine barn daily for composting. The remaining liquid part containing less nutrients is disposed through gravity to outdoor storage facilities for land application [10,11]. 

Swine manure management practices in Korea differ significantly from those of livestock farming in EU and other parts of the world, owing to the weather, geography, and limited available land. In Korea, swine manure undergoes several processes before final disposal, while in European countries, the slurry is transferred to bio-gasification facilities after a short period in the storage tank [12,13]. The floor structure of the swine barn in Korea mostly consists of the slurry pit where feces and urine, as well as washing water, are aggregated beneath the floor. The treatment of manure begins with solid−liquid separation followed by solid composting. The liquid part is treated with liquid composting or purification. Moreover, compost in the form of solids and liquids contributes approximately 85% of manure treatment, and the remaining 15% is purified and released to receiving water bodies [14]. All of these conventional modes of manure management disregard the reduction of pollutants in the slurry pit, which acts as a source of malodor. 

The bio-liquor circulation systems (BCS) moving the liquor from the bioreactor of the on-farm liquid composting facility into the slurry pit has been developed to lower the pollutant concentration of manure and its use over several years [15,16,17]. The operation of BCS is conventionally based on a fixed time and rate for circulation, which hinders the response to fluctuating loading on the bioreactor and leads to system failure or time (or energy) wastage during the treatment. Despite these systemic limitations, the attempts to optimize the operation of the bioreactor in the BCS process have not been carried out, at least on a farm scale. The optimization of BCS is significantly limited because treatment efficiency is affected by various factors such as the complex nature of manure, climate, season, microbial activity, feed, pig age, gender, slurry pit sizes, and liquid composting facilities [12]. 

In order to overcome the limitations mentioned above, as well as to contribute to improving the environment of the farm and pig productivity via the reduction of malodor, a real-time control strategy applied with a unique algorithm was designed and applied on a pilot scale of BCS and its performance was evaluated. 

## 2. Materials and Methods

### 2.1. Experimental Setup

An algorithm based on a previous lab study [12] was applied on a newly built pilot scale. The performance of the designed algorithm was evaluated using the detectability of NKP and NBP on the ORP- and pH-time profile, respectively. A new swine barn measuring 40 m^2^ and containing a slurry pit (6 m^3^) beneath the floor was constructed and connected with two tanks, designed as a bioreactor and a storage tank. The barn was fitted with a heater, ventilation fan, an automatic watering system (drinking fountain nipple), feeder, and other electrical accessories. The pigs were fed ad libitum using the automated system. The temperature of the barn was set to 27 °C, and the ventilation fans and heaters were controlled automatically (KO-880, KUNOK Co., Ltd., Nonsan, Korea).

### 2.2. Construction of Bioreactor Experimental Setup

A polyethylene tank with a capacity of 5 m^3^ was prepared using a working volume of 3 m^3^ for the bioreactor. To prevent the entrance of large solid particles during swine manure inflow from the slurry pit to the bioreactor, a 1 m^2^ sieve cradle was placed on top of the bioreactor. The settling tank was also set before setting the sieve to preclude the clogging of sieve pores due to the solid fraction of the swine manure. For the circulation of the bio-liquor, a pump was installed between the bioreactor and slurry pit. An additional pump was also installed to transfer the effluent from the bioreactor to the storage tank. The air diffuser was placed at the bottom of the bioreactor and connected with the aerator controlled by the flow meter, and a stirrer was fitted at the center of the bioreactor for the mixing of the bio-liquor (Figure 1). ORP and pH sensors were mounted for real-time monitoring of the bio-liquor in the bioreactor. Two water level sensors were installed to maintain a constant bio-liquor level in the bioreactor. All of the sensors were connected to the control panel fitted with monitoring devices. 

The experimental setup of the pilot scale BCS and the real-time control algorithm of the bioreactor were designed according to our previous study conducted with the lab-scale study [12]. 

### 2.3. Development and Evaluation of Operational Algorithm 

Figure 2 shows the flow diagram of the proposed algorithm. The operation started with the mixer-ON and the inflow of swine slurry into the bioreactor was controlled using the level sensor. An average of 60 values of ORP and pH in mV unit per minute were logged and used to calculate the moving slope change (MSC) to process the signal. The ORP_msc_ and pH(mV)_msc_ were calculated using the sample sizes of every 10 values (r = 10) per minute and were displayed as a time profile trend. Furthermore, details on the data logging and processing have been reported in past studies [18,19]. In the anoxic phase, the real-time control point (nitrate knee point, NKP) was detected by tapping the ORP_msc_ (r = 10) or the absolute value of ORP (ORP_abs_). The ORP_msc_ and ORP_abs_ were <−25 and <−240, respectively, indicating the detection of NKP and the completion of denitrification in the anoxic phase [20]. Such a trigger value was set to turn the subsequent operation based on continuous detection and following the previous lab scale study, i.e., in the anoxic phase, ORP_msc_ was below −20 for the first drop in ORP (FDO) and higher than −10 for the appearance of the plateau point (APP). In addition, ORP_msc_ was below −25 for the second drop in ORP (SDO), indicating the occurrence of NKP. Alternatively, the backup trigger value of −240 in ORP_abs_ was incorporated as a safety measure in the algorithm for the identification of NKP in case it was not detected by ORP_msc_. After acquiring the NKP, aeration was started immediately in order to lead the bioreactor operation into the aerobic phase. In the aerobic phase, the computer started the trapping of 0.4 and then −0.6 on the pH(mV)_msc_ time-profile as a trigger value for the detection of NBP. NBP describes the completion of nitrification and corresponds to the real time control point in the aerobic phase [20]. Following the appearance of NBP on the pH(mV)_msc_-time profile, the aerator and mixer were turned off and settling was maintained for 30 min followed by the discharge of the supernatant to the slurry pit for circulation or storage. Afterwards, the operations were reset and a new cycle was started with the inflow of manure to the bioreactor.

The following equations will be used for checking the system efficiency.
(1)$$\mathrm{Successful}\;\mathrm{rate}\;\left(\%\right)=\frac{\mathrm{No}.\;\mathrm{of}\;\mathrm{detected}\;\mathrm{NKP}\;\mathrm{or}\;\mathrm{NBP}\;}{\mathrm{Total}\;\mathrm{No}.\;\mathrm{of}\;\mathrm{cycle}}\times 100$$
(2)$$\mathrm{Failiure}\;\mathrm{rate}\;\left(\%\right)=\left(1-\frac{\mathrm{No}.\;\mathrm{of}\;\mathrm{detected}\;\mathrm{NKP}\;\mathrm{or}\;\mathrm{NBP}}{\mathrm{Total}\;\mathrm{No}.\;\mathrm{of}\;\mathrm{cycle}}\right)\times 100$$
(3)$${\mathrm{NKP}\;\mathrm{detected}\;\mathrm{by}\;\mathrm{ORP}}_{\mathrm{msc}}\;\;\left(\%\right)=\frac{\mathrm{No}.\;\mathrm{of}\;\mathrm{NKP}\;\mathrm{detected}\;\mathrm{by}\;{\mathrm{ORP}}_{\mathrm{msc}}}{\mathrm{Total}\;\mathrm{No}.\;\mathrm{of}\;\mathrm{cycle}\;}\times 100$$
(4)$${\mathrm{NKP}\;\mathrm{detected}\;\mathrm{by}\;\mathrm{ORP}}_{\mathrm{abs}}\;\;\left(\%\right)=\frac{\mathrm{No}.\;\mathrm{of}\;\mathrm{NKP}\;\mathrm{detected}\;\mathrm{by}\;{\mathrm{ORP}}_{\mathrm{abs}}}{\mathrm{Total}\;\mathrm{No}.\;\mathrm{of}\;\mathrm{cycle}\;}\times 100$$

### 2.4. Operational Conditions 

In this study, the slurry pit and bioreactor were initially filled with bio-liquor collected from a swine wastewater treatment process. The characteristics of the bio-liquor are presented in Table 1. The circulation between the slurry pit and the bioreactor started a week later to obtain a certain level of manure accumulation in the slurry pit. The operational conditions used in this study are concisely presented in Table 2.

The designed algorithm was employed for the real-time control of BCS on the pilot scale based on previous studies on lab scale [12,20] and preliminary studies (unpublished results). These studies showed that ORP and pH(mV) were effective real-time control factors for anoxic and aerobic operations of BCS, respectively. Meanwhile, the high inflow of solids into the bioreactor caused by improper separation of liquids/solids, owing to sieve clogging during the separation process, was identified as the main limitation of the preliminary system. Therefore, a settling tank was installed between the slurry pit and bioreactor in the pilot scale setup in order to prevent the transfer of large solid particles into the bioreactor. A complete cycle of BCS in this study comprised of the anoxic (denitrification) and aerobic (oxidation and nitrification) phases, followed by a settling step with a duration of 30 min, and was discharged either to the slurry pit or storage tank before the system was reset for the new cycle (Figure 2). 

### 2.5. Sampling

In this study, weekly sampling was performed at three sampling points of the slurry pit. The samples were vigorously mixed and a 200 mL sample was maintained below 4 °C until the required analysis. Bio-liquor and influent sampling were also carried out at the end of the circulation line and below the sieve cradle, respectively. 

### 2.6. Analytical Method

The collected samples were mixed well and then divided into two aliquots of 100 mL each and marked as original and filtered liquid samples. Then, the subsequent liquid samples were centrifuged at 3000 rpm for 10 min and were sieved using filter paper (Whatman No.6). The filtrate was stored in the refrigerator at <4 °C before the analysis and used to analyze the soluble total organic carbon (STOC), NH_4_-N, and NO_x_-N; the aliquot containing original sample (unfiltered) was used to measure the total solid (TS), total volatile solid (TVS), total suspended solid (TSS), total volatile suspended solid (TVSS), and total nitrogen (T-N). To analyze the total Kjeldahl nitrogen (TKN), the samples were digested with sulfuric acid at 380 °C for at least 4 h, and then diluted into the appropriate concentration range with distilled water [12]. The NH_4_-N, NO_x_-N, and TKN were analyzed using an auto-water analyzer (QuikChem 8500, Lachat, Milwaukee, WI, USA) equipped with the auto-dilutor. The STOC analysis was performed using an automated TOC analyzer (Torch, Teledyne Tekmar, Mason, OH, USA). All of the analyses were adherent to the standard method stipulated for water and wastewater analysis [21]. 

## 3. Results and Discussion

### 3.1. Real-Time BCS (Validation of Real-Time Control of BCS)

A total of 476 cycles were performed for 79 days and there were large variations in the cycles per day (1.5 to 13 cycles/d), as shown in Figure 3. In addition, the selected ORP and pH(mV)-time profiles, which reflect variation in the number of cycles (the numbers of cycles per day, were 10, 3, and 6 in (A), (B), and (C), respectively) are shown in Figure 4. These results demonstrated the clear performance of the real-time control technology in optimization, because manure would have been untreated or over-treated (deteriorating treatment capability and consuming energy) under a fixed time control condition. Normally, the variation in the number of cycles per day was not responded by a given operational time, but the bioreactor conditions such as the loading rate of manure and its composition, microbial activities and number, nutrient availability, temperature, and season. 

The importance of real-time control technology for BCS is well demonstrated in Table 3. The removal efficiency of NH_4_-N always reached 100%, even when the loading rate into the bioreactor and the F/M ratio were changed, because of the variations in the composition of manure in the slurry pit. These results should indicate that the control technology based on comprehending the ORP and pH(mV) time-profile can present a way of controlling and optimizing the process in real time. However, the diagnosis and optimization of the conditions of the bioreactor in fixed time control using the traditional method is impossible. During the operation of the bioreactor with the designed real-time control technology, the operational sequences of a bioreactor were performed with a daily average of 6.03 cycles, and 100% NH_4_-N removal was guaranteed. The obtained average loading rate and F/M ratio based on NH_4_-N were 82 g/m^3^·d and 0.018, respectively. The average soluble nitrogen (NH_4_-N + NO_x_-N) removal efficiency was 98% and the average NOx-N concentration in the effluent was 6.4 mg/L.

### 3.2. ORP and pH (mV)- Time Profile

The relationship between the morphological changes of nitrogen and ORP and pH (mv)-time profiles in the bioreactor was tracked to determine whether the bioreactor was adequately controlled by the designed algorithm (Figure 5). The devised algorithm was found to control the process by stably detecting the timing of the complete removal of NH_4_-N and NO_x_-N in the bioreactor, while diagnosing changes in the ORP and pH-time profiles. As the influent from the slurry pit was loaded into the bioreactor, the concentration of NH_4_-N increased gradually during the anoxic phase via the reduction of NO_x_-N. The NO_x_-N concentration reached 0 at 5 and 12.2 h through denitrification using organic matter in the influent. With the influent loading in the anoxic phase, the ORP curve fell rapidly and was maintained at a constant level or was decreased slightly. Furthermore, it decreased sharply at the point of complete denitrification (NKP). In contrast, no specific changes indicating the completion of denitrification were observed in the pH(mV)-time profile, and some noise signal and unstable trend appeared in the pH(mV) curve. This observation might reveal the reason the pH(mV)-time profile was not considered as a real-time control factor for the anoxic phase in this study. For reference, the typical trend of pH(mV) decrease during denitrification was likely due to alkalinity release; notably, pH and pH(mV) were antonymous [19].

Following the detection of the end point of denitrification on the ORP time-profile, nitrification reaction actively occurred in the aerobic condition, obtaining complete nitrification at 6 and 13.8 h. At this point, the pH(mV) was observed to sink suddenly. After identifying this singularity on the pH(mV)-time profile by the designed algorithm, settling and influent loading were successively performed. Overall, the ORP-time profile allowed for the accurate detection of the denitrification termination point (NKP). Meanwhile, the pH(mV)-time profile ensured the accurate detection of the nitrification termination point (NBP), despite some noise signal in the aerobic phase, indicating that ORP and pH(mV) as single control factors for the anoxic and aerobic phases, respectively, were adequate for the efficient and stable operation of BCS. The relationship between ORP and pH time-profiles and nitrogen changes in the bioreactor have already been published and more detailed knowledge can be obtained from [12,18,20].

Figure 6 presents ORP and pH(mV)-time profiles for the comprehensive analysis of the entire process. Figure 6A shows that NKP was detected in the ORP-time profile in the first cycle in an excellent manner, and this plot shows all three steps (i.e., FDO, APP, and SDO) before entering the aerobic phase. Figure 6A illustrates that the decline in pH(mV) stopped for a short period upon acquiring NKP. This indicated the completion of denitrification and the elimination of the alkalinity release. Similarly, in the second cycle after the identification of NBP on the pH(mV)-time profile in the aerobic phase, followed by 30 min settling time, the anoxic phase started with the inflow of manure. In the anoxic phase, FDO appeared instantly, followed by SDO; however, APP and, subsequently, NKP were not detected, thereby causing a system shift to the ORP_abs_-base control (trigger value −240 mV) for the detection of NKP_sup_ (SDO_sup_). As mentioned earlier, a trigger value of −240 mV for ORP_abs_ was set as a back-up for ORP_msc_(mV). This back-up value ensured the system does not stop in the anoxic phase when the ORP_msc_ fails to detect NKP. In contrast, the dropping of the ORP and tapping of the ORP_abs_ trigger value showed a flat trend on the pH(mV) scale, which indicated the completion of the anoxic period (denitrification) at SDO. Furthermore, it created an anaerobic condition that supported anaerobes, in order to produce organic acid and stop alkalinity release before the beginning of the aerobic phase. This phenomenon was also reported by Won and Ra (2011) [20]. 

Additional details on this phenomenon are shown in Figure 6B, which provides a clear illustration of FDO and SDO, while missing the clear APP in both cycles. It steered the switch of the ORP_msc_ to ORP_abs_ trigger for the detection of NKP_sup_ (SDO_sup_), and subsequently entered the aerobic phase (nitrification stage). The undetected NKP by ORP_msc_ might be due to the variable composition of the slurry, which includes a high concentration of organics that promote fast denitrification in the bioreactor in the anoxic phase.

In Figure 6C, an unusual trend of pH(mV) was observed; i.e., when the phase changed from anoxic to aerobic after the second NKP detection, pH(mV) exhibited continuous reduction rather than an increasing trend. This indicates that no nitrification was carried out owing to the unavailability of NH_4_-N, which was the outcome of no influent feeding due to the breakdown of the influent pump. Moreover, the reduction in pH(mV) in the aerobic phase was associated with CO_2_ striping, as reported previously [12], [20]. This CO_2_ stripping depended on the CO_2_ concentration in the medium and possibly reduced the steepness of the pH(mV) over time [22]. In contrast, NKP detected even the breakdown of the pump, which was due to the presence of some carbon in the bioreactor in order to support denitrification. Therefore, because influent feeding was hindered by the malfunction of the influent pump, NBP on the pH(mV)-time profile did not occur in the aerobic phase, resulting in the continuous maintenance of aerobic conditions. Therefore, the system was manually reset for influent feeding, which guided the resumption of normal real-time BCS working. The breakdown of the influent pump occurred twice (0.4%) during the study, resulting in a 99.6% success rate (error free) and reliably high efficiency of real-time BCS. 

### 3.3. Quantitative Analysis of NKP Detection by ORPmsc and ORPabs and Its Impact on Cycle Characteristics

Table 4 represents the quantitative information of NKP detection by ORP_msc_ and ORP_abs_ in the anoxic phase. In this study, three types of real-time control patterns were observed in a single day. These include the spate NKP detection by ORP_msc_ and ORP_abs_; both share NKP detection either by ORP_msc_ or ORP_abs_. The number of cycles for NKP detection by ORP_msc_ only, both (ORP_msc_ and ORP_abs_), and ORP_abs_ only was 291 (56 days), 132 (16 days), and 53 cycles (7 days), respectively. The detection of NKP is normally associated with denitrification, which highly depends on the availability of organic matter [23,24,25]. As no external carbon source was used to enhance the denitrification, the difference in the number of cycles per day depended on a number of factors, such as organic matter availability, microorganism activities, and other environmental factors [12,23].

The entire detection of NKP was perfectly conducted with 78.6% by ORP_msc_ and 21% by ORP_abs_, except 0.4% for the malfunction of the influent pump, showing that NKP detection by ORP_msc_ was more frequent than that by ORP_abs_. The control by ORP_msc_ constantly allowed for the immediate start of aeration after detecting NKP. In contrast, control by ORP_abs_ required an interval of approximately 28 min on average from NKP to the beginning of aeration.

### 3.4. Changes of Swine Manure Characteristics in the Slurry Pit 

The average concentration values of NH_4_-N in the slurry pit were calculated to be 418.4 ± 67.7 and 1624.5 ± 244.0 mg/L in the bio-liquor circulation system (BCS) and non-bio-liquor circulation system (NBCS), respectively, showing a 74.2% reduction of NH_4_-N in BCS (Table 5). The STOC concentration also showed a similar reduction in the efficiency, with 1791.2 ± 21.3 and 7406.5 ± 2181.9 mg/L in BCS and NBCS, respectively. Additionally, the reductions in the concentration of solid fractions such as TS, TVS, TSS, and TVSS were 72.0%, 75.0%, 75.3%, and 74.8%, respectively.

Because the reduction in NH_4_-N can subsequently reduce the generation of harmful gases, such as NH_3_, in the slurry pit, as reported in recent studies [26,27,28,29,30], in order to understand the NH_3_ emission reduction effect of the real-time controlled BCS, the theoretical NH_3_ gas mass generated from each slurry pit was estimated using the NH_4_-N concentration and pH data, and the NH_3_ concentration inside the pig sty was monitored. Therefore, the theoretical NH_3_ generation values of NBCS and BCS were 21.6 and 12.7 mg/m^3^·d, respectively, which was approximately 41.0% lower in BCS, and the monitored NH_3_ concentration also averaged 14.6 ± 3.4 ppm in NBCS and 5.6 ± 1.6 ppm in BCS, yielding a 61.6% reduction in BCS. These results could indicate that the real-time controlled BCS developed through this study could be a very effective technology for pig manure treatment and a preemptive control method for reducing odor-causing substances in pigsties.

## 4. Conclusions

The improvement of pigsty environments and manure treatment is essential for in-creasing pig productivity, reducing odor complaints, ensuring the safety of field employees, and preventing environmental pollution. Among the livestock species in Korea, most civil complaints of malodor occurs from the vicinity of swine farms. Although the concept of BCS may be a good system to remove the origin of malodor, its performance has not been guaranteed with the designated control strategies under the necessary time frame. In general, it has been well known that optimized operation is impossible in a conventional system controlled in fixed-time mode because various factors affect the operational efficiency of the bioreactor. This study shows the elimination of odor-causing substances via the real time control technology for BCS operation. The real-time control of the bioreactor using a well-designed algorithm always had an optimized manure treatment performance, regardless of the influent concentration, and thus is expected to greatly contribute to the improvement in the operating efficiency of the BCS. In this study, the concentration of NH_3_ in the pigsty of the BCS controlled in real time was only 5.6 ppm, which was 61.6% lower than that of the conventional system. Therefore, it is considered that the management of manure in a slurry pit using real-time BCS is effective at reducing odor. As a result of the experiment, all real-time operations were successfully performed, but twice (0.4%), control failures occurred due to influent pump trouble. Nevertheless, the optimized operation of the bioreactor using the ORP and pH time-profiles will reduce the management labor of BCS and greatly increase the environmental improvement effect of the swine farm. This study may provide reference for improving the BCS performance and pigsty environments with relatively little labor, and provide further guidelines for the optimization of manure treatment in the bioreactor.

## Figures and Tables

**Figure 1 animals-12-02941-f001:**
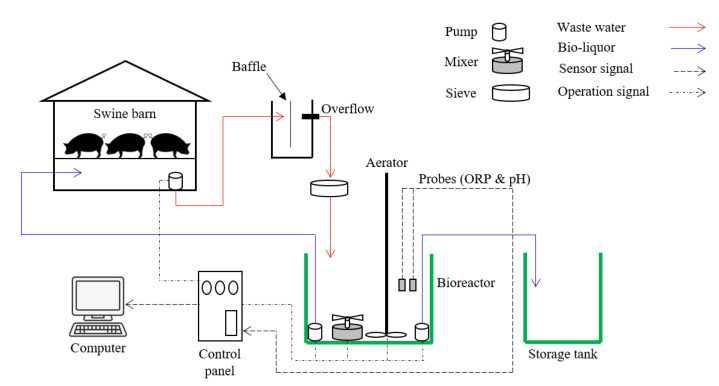
Bio-liquor circulation system setup.

**Figure 2 animals-12-02941-f002:**
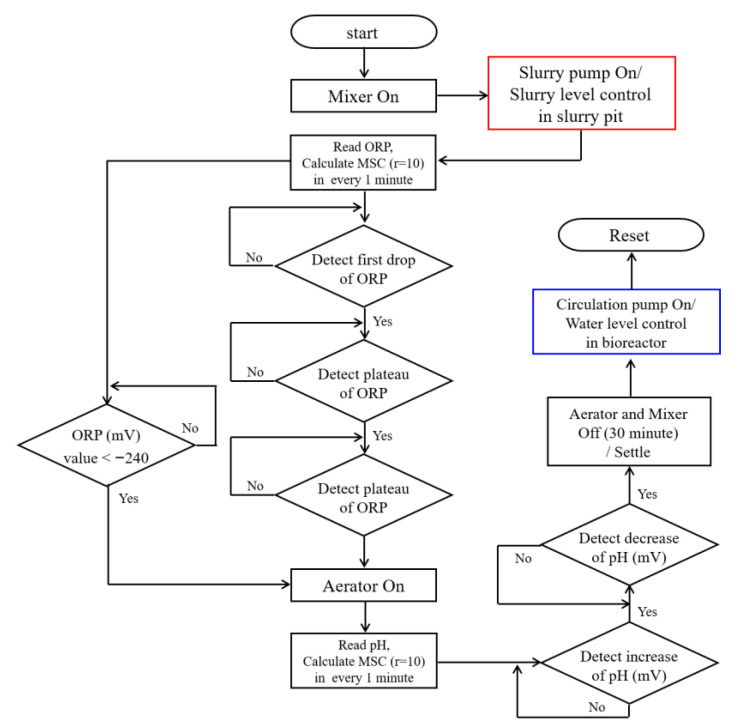
Operational algorithm for real-time control of bio-liquor circulation system.

**Figure 3 animals-12-02941-f003:**
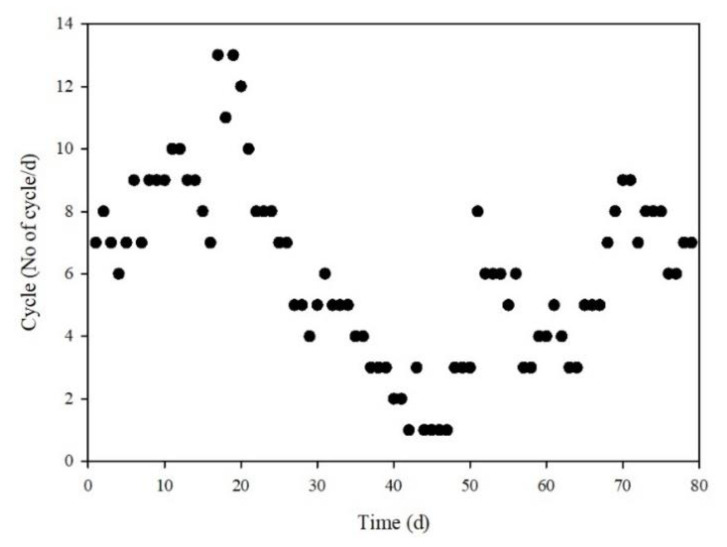
Variation in no. of cycles per day during operation.

**Figure 4 animals-12-02941-f004:**
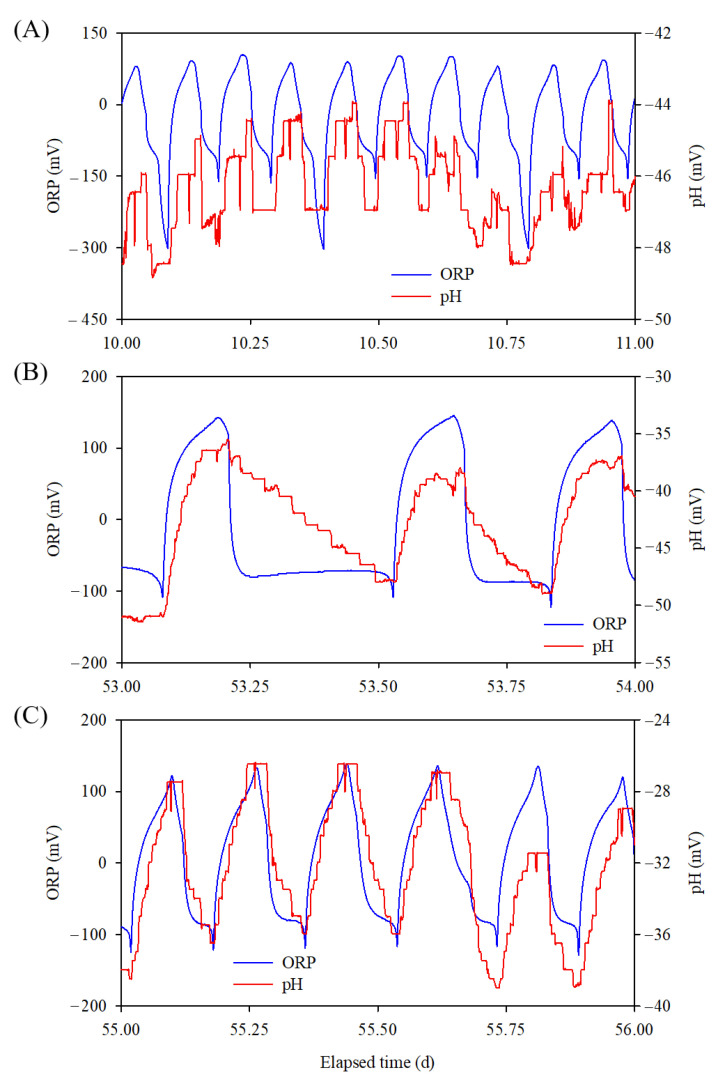
Typical trends selected for the algorithm validation: The automatically changed number of runs per day, (**A**) 10 runs/day, (**B**) 3 runs/day, (**C**) 6 runs/day.

**Figure 5 animals-12-02941-f005:**
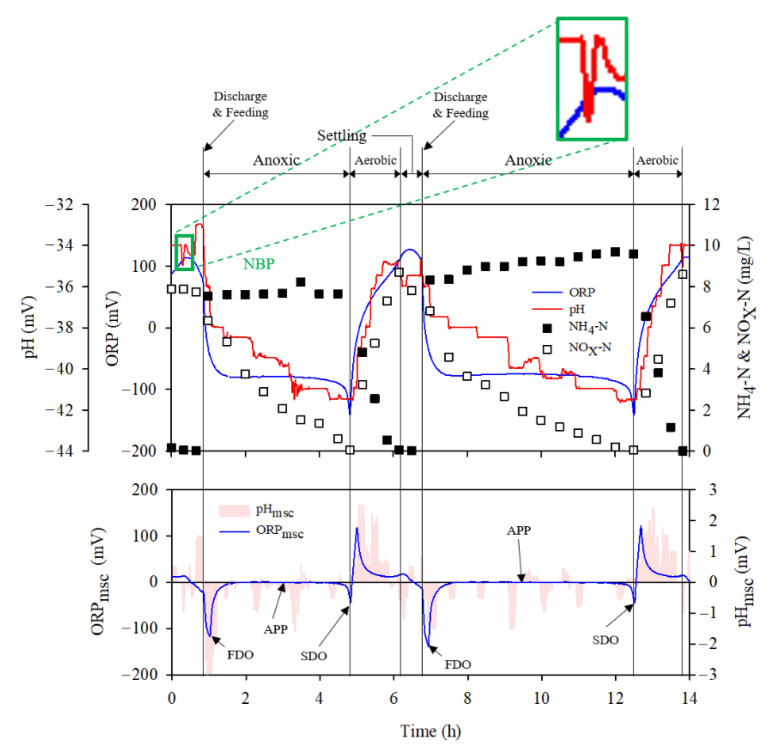
Representative ORP and pH(mV) time-profile: Upper, the blue and red lines show the time-profile of ORP and pH, respectively. Dark and light squares show the concentration of NH_4_-N and NO_X_-N, respectively. Lower, pink area and blue line show the pH_msc_ and ORP_msc_, respectively.

**Figure 6 animals-12-02941-f006:**
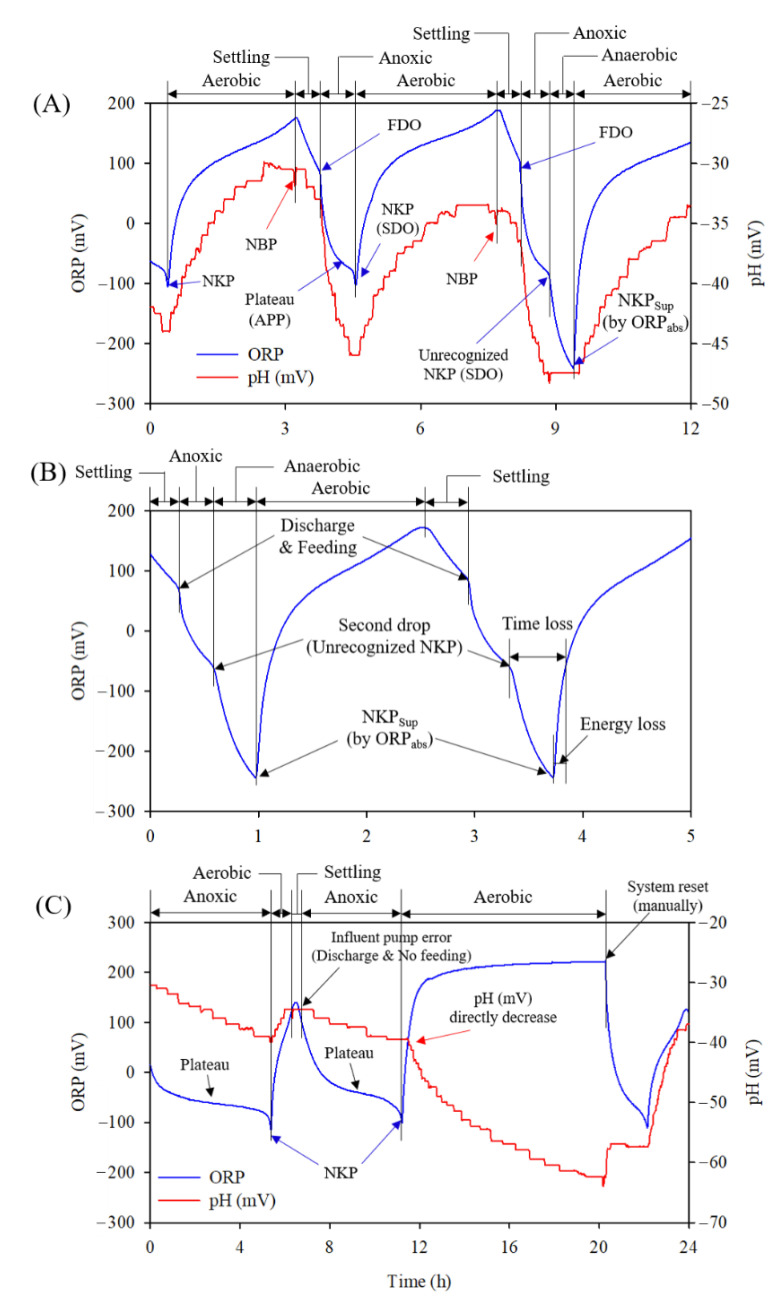
Comprehensive analysis of ORP and pH(mV) profiles obtained during the real-time of BCS: (**A**) NKP & ORP_abs_ for denitrification detection, (**B**) ORP_abs_ for denitrification detection, (**C**) Malfunctioning pump.

**Table 1 animals-12-02941-t001:** Initial characteristics of bio-liquor (activated sludge) used for preliminary and pilot scale study (unit: mg/L).

Parameters	Value
TS *	8000.0 ± 57.7
TVS	4733.3 ± 230.9
MLSS	6433.3 ± 208.2
MLVSS	3633.3 ± 176.4
STOC	568.8 ± 4.4
NH_4_-N	1.2 ± 0.1
NO_X_-N	0.9 ± 0.0
TKN	624.4 ± 3.2
T-N	625.3 ± 4.6

* TS, total solids; TVS, total volatile solids; TSS, total suspended solids; TVSS, total volatile suspended solids; STOC, soluble total organic carbon; TKN, total Kjeldahl nitrogen; T-N, total nitrogen.

**Table 2 animals-12-02941-t002:** Operational conditions of the bioreactor and slurry pit.

Operational Factors	Real Time-Controlled Bio-Liquor Circulation System		Non-Bio-Liquor Circulation System Slurry Pit
	Bioreactor	Slurry pit	
Working volume (m^3^)	3	6	6
Initial condition	Filled with bio-liquor	Filled with bio-liquor	Empty
Default circulation rate (m^3^/cycle)	0.12	0.12	
Circulation rate based on volume (%/cycle)	4	2	-
Aeration rate (m^3^/m^3^·min)	0.05	-	-

**Table 3 animals-12-02941-t003:** Bioreactor operation results and removal efficiency.

Avg. (Cycles/d)	Avg. Loading Rate Based on NH_4_-N (g/m^3^·d)	Avg. F/M * (ratio/d)	Influent (mg/L)		Effluent (mg/L)		Removal (%)	
			NH_4_-N	NO_X_-N	NH_4_-N	NO_X_-N	NH_4_-N	NH_4_-N & NO_X_-N
6.03 ± 2.82	81.9 ± 40.8	0.0177 ± 0.0431	352.8 ± 75.7	0.0 ± 0.0	0.0 ± 0.0	6.4 ± 3.0	100 ± 0.0	98.1 ± 1.0

* Avg. F/M, Avg. mg NH_4_-N/mg MLVSS d.

**Table 4 animals-12-02941-t004:** Quantitative analysis of ORP_msc_ and ORP_abs_ for NKP detection on the cycle characteristics.

Parameter	ORP_msc_ (Alone)	ORP_abs_ (Alone)	Both (ORP_msc_ and ORP_abs_)
No of Days	56.0	7.0	16.0
No of Cycles	291.0	53.0	132 (84 ORP_msc_, 48 ORP_abs_)
Percentage (%)	78.78	21.21	-

**Table 5 animals-12-02941-t005:** Comparison of the final characteristics of the swine manure in a slurry pit of a real-time control bio-liquor circulation system and conventional slurry pit system.

Parameter (mg/L)		Conventional Slurry Pit System, NBCS ^(1)^	Real Time-Controlled Bio-Liquor Circulation System, BCS ^(2)^	Reduction Efficiency (%)
NH_4_- N		1624.5 ± 244.0	418.4 ± 67.7	74.2
NO_X_-N		ND ^(3)^	ND	-
STOC		7406.5 ± 5181.9	1791.2 ± 521.3	75.8
TKN		4396.7 ± 2090.2	1579.1 ± 722.8	64.1
TS		65,839.3 ± 31,862.7	18,462.7 ± 6991.3	72.0
TVS		48,795.6 ± 24,968.3	12,192.5 ± 5270.0	75.0
TSS		59,427.8 ± 31,002.7	14,694.0 ± 6868.4	75.3
TVSS		44,912.7 ± 24,079.9	11,333.3 ± 5349.3	74.8
pH		7.25 ± 0.30	7.64 ± 0.21	-
Theoretical NH_3_ emission (mg/m^3^·d)		21.6	12.7	41.0
NH_3_ concentration in swine barn (ppm)	Avg.	14.6 ± 3.4	5.6 ± 1.6	61.6
	Max.	25.6	15.4	39.8
	Min.	6.7	3.1	53.7

^(1)^ NBCS, non-bio-liquor circulation to slurry pit; ^(2)^ BCS, bio-liquor circulation to slurry pit; ^(3)^ ND, not detected.

## Data Availability

The data presented in this study are available on request from the corresponding author.

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
