# Peer review of "Real-Time Control Technology for a Bio-Liquor Circulation System in a Swine Barn with Slurry Pit: Pilot Scale Study"

_animals, 2022, doi:10.3390/ani12212941_

Round 1

Reviewer 1 Report

An effective real time control technology for the bio-liquor circulation system was discussed in this manuscript. The duration of the anoxic and aerobic phases could be automatically optimized using real-time control technology, and the nutrients concentration in the manure of the slurry-pit could be noticeably lowered, which showed the real time control technology had certain guideline significance to engineering practice of manure treatment in the bioreactor. The manuscript could be improved in these aspects:

Line 36:  “bio-liquor” was not suitable to be the keyword

 Line 107-109: “An additional pump was installed on the opposite side of the circulation pump to transfer the effluent from the bioreactor to the storage tank.” The sentence should be modified.

 Fig. 1: The line of “storage tank” and “bioreactor” should be bold.

 Table 1: The characteristics of bio-liquor were usually expressed in MLSS and MLVSS, why “TS, TVS, TVSS, and TSS” were used here? The value of “TVSS” was less than “TVS”. Is that right?

 Table 2: “0.12” was not suitable to be put in the middle.

 Table 3 is missing.

 Line 204-205: NO. of the cycles per day changed during operation, more explaining should be provided.

 Fig.5: More detailed information of Figure should be provided.

 Line 257: “took 6 and 13.8h” what does this mean in the sentence?

 Line 336-337: “72.0, 75.0, 75.3” should be “72.0%, 75.0%, 75.3%

 Line 355: “sties”→“sty”?

Author Response

Thank you for good comment.

Reviewer 2 Report

Reviewer Comments:

Highlights: Not provided.

Graphical Abstract: Not provided.

General comment:

This paper is study about a real time control strategy with a unique algorithm was designed and applied on a pilot scale of BCS, and its performance was evaluated. Overall, this paper has well covered the authors’ findings on a bio-reactor that was operated in real-time manner based on diagnosis of ORP and pH(mV) time-profiles in a pilot scale. However, authors need to perform critical analysis and interpret all these studies and come up with a conclusion for each section. It’s good that the author had this finding written but readers would preferably want to know what had the authors concluded from all these studies, instead of what the author of the literature studies had concluded. To conclude, this paper needs to revise it carefully before it can be considered in high impact journal. Hope below comments will be able to help to further improve the paper.

Specific comment:

Abstract:

Kindly include the problem statement of this study in the abstract as the abstract should be able to stand alone

Please include a conclusion to wrap up the abstract at the end, the current abstract ended too suddenly

Introduction:

Introduction should be covered the gap of the research. However, it is not well covered in this section.

Also, please mention the important of this study to society as well as industry.

Kindly compare the advantages between the proposed method and the current methods used to treat manure

Kindly include a summary of novelty of this study at the last paragraph of the introduction

It is good to describe more on the proposed method, such as any previous studies that used similar concepts, working principle of this method etc.

Kindly refer some latest papers as it is highly relevant to this report. Example, “Optimization of Hydrolysis-Acidogenesis Phase of Swine Manure for Biogas Production Using Two-Stage Anaerobic Fermentation” and “Optimizing real swine wastewater treatment efficiency and carbohydrate productivity of newly microalga Chlamydomonas sp. QWY37 used for cell-displayed bioethanol production”

Material and Method:

Kindly refrain from using the term “our previous study”, change it to passive form instead

Figure 2 can be improved, some boxes are too small, also kindly check all words used in the figure

Body:

Results

Kindly improve on the discussion. What is the significance of the results of the work?

Please give more authors’ opinion before including a literature reference as support, what does the authors think of the results and why does the results act so? After mentioning these then only place the citations to support the authors’ claims.

Kindly standardize whether to bold the text “Figure X” in text, some are in bold while others are not, please revise

After placing a citation to support the claims made in the discussion, authors are encouraged to further elaborate the point to further convince the readers

Table 5 was not mentioned in text, kindly revise

There is no “discussion” section, kindly double check the authors’ guidelines and see if it is required. Authors can also consider renaming this section as “results and discussion”  

The authors are encouraged to read this article for more scientific information, “Sustainable Smart Photobioreactor for Continuous Cultivation of Microalgae Embedded with Internet of Things”

Conclusions

Kindly finish the conclusion in one paragraph

Please include the problem statement in the conclusion as a conclusion should be able to stand alone

The conclusion does not properly wrap up the study, the current conclusion sounds more like repeating all the results reported, kindly revise the whole conclusion

References

Kindly revise reference format according to the author guideline.

It is suggested to cite references within 5 years of research to maintain the reliability of results obtained.

There are references found to be outdated.

Additional materials for reading:

“Occurrence and transfer characteristics of blaCTX-M genes among Escherichia coli in anaerobic digestion systems treating swine waste” and “Impact of influent strengths on nitrous oxide emission and its molecular mechanism in constructed wetlands treating swine wastewater”

Author Response

Thank you for good comments.

Round 2

Reviewer 2 Report

I have gone through the revised manuscript and I am happy with the changes made.